# A CRISPR-Cas12a-based universal rapid scrub typhus diagnostic method targeting 16S rRNA of *Orientia tsutsugamushi*

**Bum Ju Park[1,2], Sang Taek Heo[3], Misun Kim[3], Jeong Rae Yoo[3], Eun Jin Bae[4], Su Yeon Kang[4], Sunghoon Park[5], Kyeo Re Han[5], Keun Hwa Lee[4], Jae Myun Lee[5], Hyeyoung Lee[2,6], Yoon-Jae Song** [1] *

**1** Department of Life Science, Gachon University, Seongnam-Si, Republic of Korea, **2** Inogenix Inc. Gangwon-Do, Republic of Korea, **3** Department of Internal Medicine, Jeju National University College of Medicine, Jeju, Republic of Korea, **4** Department of Microbiology, Hanyang University College of Medicine, Seoul, Republic of Korea, **5** Department of Microbiology and Immunology, Institute of Immunology and Immunological Diseases, Brain Korea 21 PLUS Project for Medical Science, Yonsei University College of Medicine, Seoul, Republic of Korea, **6** Department of Biomedical Laboratory Science, College of Health Sciences, Yonsei University, Wonju, Republic of Korea

* songyj@gachon.ac.kr

**Data Availability Statement:** The manuscript provides patient demographics and clinical characteristics, such as age, gender, immunosuppression status, Charlson Comorbidity

## Abstract

Scrub typhus is caused by *Orientia tsutsugamushi* infection and occurs frequently in an area called the Tsutsugamushi Triangle. Currently, there is no vaccine for *O. tsutsugamushi*, and its infection is treated with antibiotics such as doxycycline. Scrub typhus responds to effective treatment, and early treatment shortens the course of the disease, reduces mortality, and accelerates recovery. Therefore, it is important to rapidly diagnose *O. tsutsugamushi* infection to ensure successful outcomes. Here, we developed a CRISPR-Cas12a-based diagnostic method targeting the bacterial 16S rRNA to detect *O. tsutsugamushi* infection of all known genotypes. To reduce the possibility of contamination and increase field applicability, we designed the one-pot assay system in addition to conventional two-pot assay system. Using this method, we successfully detected up to 100 copies of *in vitro* transcribed *O. tsutsugamushi* 16S rRNA within 1 hour under isothermal conditions. In blood samples from patients confirmed to be infected with *O. tsutsugamushi* by nested PCR, the developed method exhibited a clinical sensitivity of 98% and high specificity. These data demonstrate that the presented method is applicable for the rapid and universal diagnosis of scrub typhus to facilitate timely and appropriate treatment.

## Author summary

Early treatment with antibiotics can shorten the course of scrub typhus, reduce mortality, and accelerate recovery, making the rapid diagnosis of *O. tsutsugamushi* infection critically important. In this study, we developed a CRISPR-Cas12a-based rapid diagnostic method targeting the bacterial 16S rRNA to detect *O. tsutsugamushi* infection across all known genotypes. Using this method, we successfully detected as few as 100 copies of in

Index (CCI), and days since illness onset. Beyond this information, the disclosure of raw clinical data (including personal details) to third parties without prior consent from the data subjects is prohibited by law. For inquiries regarding the raw clinical data, please contact Jeju National University Hospital Institutional Review Board (irb1503@jejunuh.co.kr). The utilized primer sets, designed to detect the OT1 and OT2 sequences (Table 1), were designed according to the provided protocol, with reference to the aligned 16S rRNA sequence for nine genotypes of O. tsutsugamushi (S1 Fig): Karp, Kuroki, TA763, Gilliam, Kawasaki, Japanese Gilliam, Kato, and Shimokoshi (GenBank accession numbers: NR_025860.1, D38626.1, NZ_OUNA00000000.0, D38622.1, D38625.1, AP008981.1, NZ_LS398550.1, D38627.1, respectively; NCBI). The Saitama strain was excluded because it was not published in NCBI.

**Funding:** This research was supported by a grant from the Korea Health Technology R&D Project through the Korea Health Industry Development Institute (KHIDI), funded by the Ministry of Health & Welfare, Republic of Korea (HI22C0286). The funder had no role in study design, data collection and analysis, decision to publish, or preparation of the manuscript.

**Competing interests:** The authors have declared that no competing interests exist.

vitro-transcribed *O. tsutsugamushi* 16S rRNA within 1 hour under isothermal conditions. In blood samples from patients confirmed to be infected with *O. tsutsugamushi* by nested PCR, the developed method demonstrated a clinical sensitivity of 98% and high specificity.

## Introduction

Scrub typhus, caused by *Orientia tsutsugamushi* (*O. tsutsugamushi*), is a clinically significant vector-borne disease prevalent in the Asia-Pacific region, particularly within the Tsutsugamushi Triangle, which encompasses the Russian Far East, northern Australia, Afghanistan, and surrounding areas. Transmission occurs via the bite of infected Trombiculidae larvae, with clinical symptoms including fever, rash, and eschar at the bite site [1]. Severe cases may lead to complications such as hemophagocytic syndrome, meningitis, and multi-organ failure, with a mortality rate of approximately 6% without treatment [1–4]. Appropriate antibiotic therapy, including doxycycline, tetracycline, chloramphenicol, and azithromycin, reduces the mortality rate to 1.4% [1,2,5,6]. Despite its clinical significance, there is no effective vaccine for scrub typhus, making early and accurate diagnosis essential to reducing mortality and improving patient outcomes [7].

Currently, culture tests, antibody-detection tests, and nucleic acid-based tests (NATs) are used to diagnose *O. tsutsugamushi* [8–12]. Culture tests take several weeks to perform, have a sensitivity < 50%, and must be performed in a BSL-3 laboratory; thus, this strategy is more often used for bacterial isolation than for diagnosis [13]. The antibody-detection tests include those based on an indirect immunofluorescent-antibody test (IFA), passive hemagglutination assay (PHA), and enzyme-linked immunosorbent assay (ELISA) [8,10,14]. IFA can be used to compare the antibody titer between the acute and convalescent phases, and is widely used for diagnostic purposes due to its high sensitivity and specificity [8]. However, it requires expensive equipment, such as a fluorescence microscope, and cannot be scaled up for high-throughput diagnosis. Given this, IFA is most commonly used as a confirmation test in current practice [15]. ELISA is also used to compare antibody titers for diagnostic purposes; it is less costly than IFA while yielding a similar sensitivity [14]. However, both of these techniques require comparison of antibody values in the acute and convalescent phases, and thus are not easily applicable for early diagnosis. PHA enables the user to visually detect the degree to which a patient's serum aggregates with the antigen, but its sensitivity is low [10]. NATs using real-time or nested PCR represent the current standard for diagnosing *O. tsutsugamushi* infection; it and IFA are the most widely used methods for diagnosing scrub typhus [9,11]. However, real-time PCR may return false positives due to non-specific amplification and nested PCR, which avoids false positives, requires a secondary amplification step.

*O. tsutsugamushi* is an obligate, intracellular bacterium which belongs to family *Rickettsiaceae* and causes scrub typhus. Although the 16S rRNA sequence of *O. tsutsugamushi* matches those of *Rickettsia* species of the same family by about 90% [16–18], major differences are seen in their cell wall components. The cell walls of *Rickettsia* comprise lipopolysaccharides, outer membrane B, and a 17-kDa lipoprotein, whereas those of *O. tsutsugamushi* lack these components and instead have a 56 kDa type-specific antigen (TSA56) that serves as a criterion for the diversity of *O. tsutsugamushi* strains [19, 20]. *O. tsutsugamushi* was initially classified into three prototypes (Karp, Kato, and Gilliam) based on antigenic similarity detected by complement-fixation (CF) analysis [21, 22]. Subsequently, the TSA56, which exhibits high antigenic variability, was used as a major antigenic determinant. Based on sequence analysis of the

TSA56, researchers proposed nine distinct genotypes of *O. tsutsugamushi* in 2009, namely Karp, Kuroki, TA763, Gilliam, Kawasaki, Japanese Gilliam, Kato, Saitama, and Shimokoshi [23].

Recently, a field-applicable rapid molecular method for diagnosing pathogenic infection was developed based on the clustered regularly interspaced short palindromic repeat (CRISPR) and CRISPR-associated proteins (Cas) immune system of bacteria [24–26]. The CRISPR-Cas system uses an RNA-guided endonuclease to provide sequence-specific immunity against an invaded nucleic acid sequence (e.g., from a plasmid or bacteriophage). The RNA used for the RNA-guided endonuclease is called guide RNA (gRNA) or CRISPR RNA (crRNA) and is derived from the invaded nucleic acid sequence. The previously reported diagnostic method utilized a Cas12a protein comprising a gRNA complementary to the target DNA sequence plus a T nucleotide-rich protospacer-adjacent motif (PAM). Binding of Cas12a to a target DNA activated its trans-cleavage activity to cleave nearby single-strand DNA. Using Cas12a and isothermal amplification, Jenifer Doudna and colleagues developed a molecular diagnostics technology called DNA Endonuclease-TargEted CRISPR Trans Reporter (DETECTR) [24]. Various pathogens, including human papilloma viruses (HPV), severe acute respiratory syndrome corona virus 2 (SARS-CoV-2), severe fever with thrombocytopenia syndrome virus (SFTSV), and human immunodeficiency virus (HIV) were detected using the DETECTR platform, and Cas12a-based pathogen diagnosis is currently utilized in the clinic [24,27,28].

In the present study, we designed a primer-gRNA set for the targeted detection of the 16S rRNAs of *O. tsutsugamushi*. The 16S rRNA sequence is commonly used for bacterial classification. Unlike other mRNA molecules, this rRNA resembles DNA in having a stable structure with many secondary structures, and thus is not easily degraded during its isolation from cells or use in experiments. Furthermore, whereas *O. tsutsugamushi* contains a single copy of a given DNA target [29], it may have hundreds to thousands of copies of an RNA target. The *O. tsutsugamushi* 16S rRNA is transcribed consistently enough to be used as a housekeeping gene in the microarray analysis [30]. For these reasons, we hypothesized that using the 16S rRNA gene as a detection target would yield high-level sensitivity for *O. tsutsugamushi*. Both antibody-detection tests and NATs for scrub typhus diagnosis rely on the TSA56. However, the high antigenic variability of TSA56 complicates these methods, resulting in inconsistent detection and reduced sensitivity [31]. Molecular diagnostics targeting the conserved 16S rRNA sequence address these limitations, enabling consistent and reliable detection across diverse *O. tsutsugamushi* genotypes. We designed primer-gRNA sets to detect nine genotypes of *O. tsutsugamushi*, and used them to develop a CRISPR-Cas12a-based diagnostic method for detecting *O. tsutsugamushi*, which we named *O. tsutsugamushi* DETECTR or OT DETECTR.

## Materials and methods

### Ethics statement

This study was approved by the Institutional Review Board (IRB) of Jeju National University Hospital (IRB no.2022-05-002). Written informed consent was obtained from all the participants.

### Two-pot OT DETECTR

In our two-pot Cas12a-based diagnostic method, the isothermal amplification and Cas12a trans-cleavage steps were performed in different Eppendorf tubes (two-pot) at two different temperatures (Fig 1A). The bacterial DNA and RNA were amplified by recombinase polymerase amplification (RPA) and reverse transcription recombinase polymerase amplification (RT-RPA), respectively, using a TwistAmp Basic kit (TwistDx, Cambridge, UK). The utilized

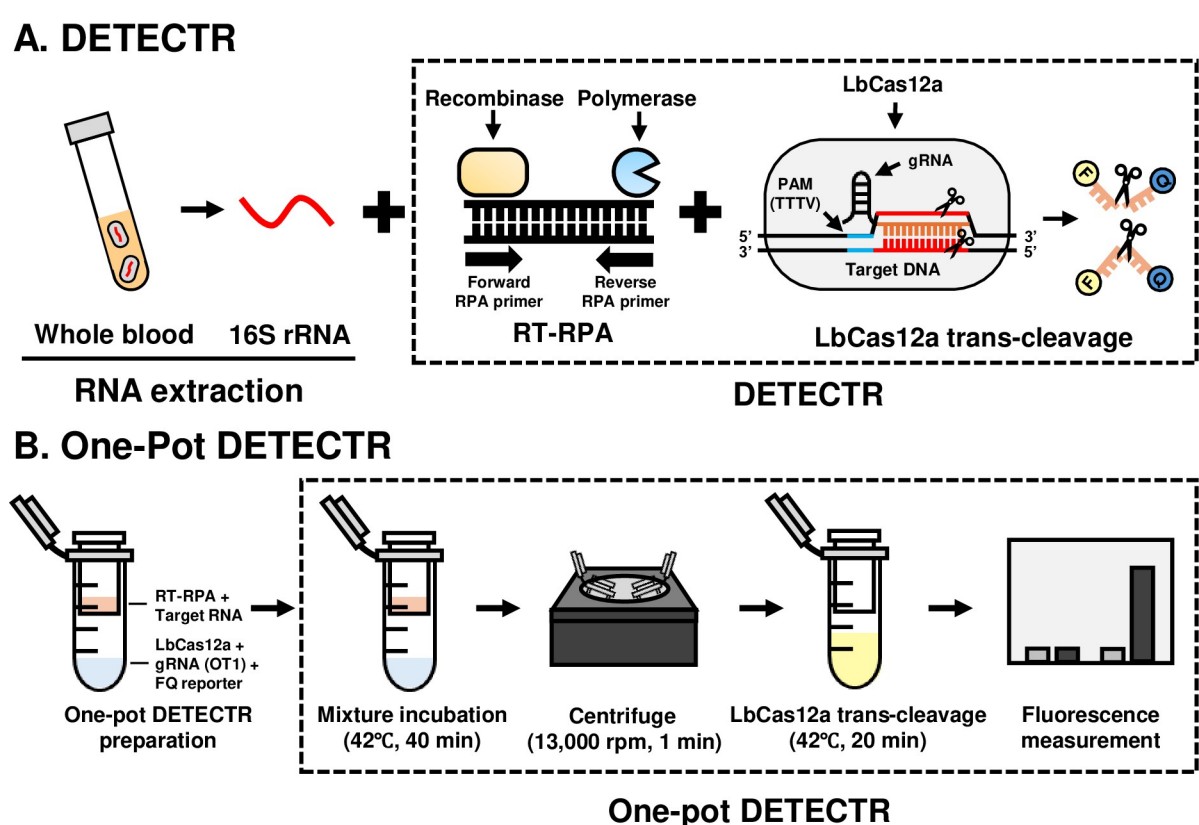

**Fig 1. Schematic diagrams of OT DETECTRs.** (A) Conventional two-pot OT DETECTR. (B) One-pot OT DETECTR.

primer sets, designed to detect the OT1 and OT2 sequences (Table 1), were designed according to the provided protocol, with reference to the aligned 16S rRNA sequence for nine genotypes of *O. tsutsugamushi* (S1 Fig): Karp, Kuroki, TA763, Gilliam, Kawasaki, Japanese Gilliam, Kato, and Shimokoshi (GenBank accession numbers: NR_025860.1, D38626.1, NZ_OUNA00000000.1, D38622.1, D38625.1, AP008981.1, NZ_LS398550.1, D38627.1, respectively; NCBI). The Saitama strain was excluded because it was not published in NCBI. As described in the provided protocol, the RPA reaction mixture comprised 29.5 μL rehydration buffer, 2.4 μL forward and reverse primers (10 μM each), and 2.5 μL of 280 mM magnesium acetate, with samples and nuclease-free water added obtain a final volume of 50 μL. Each RPA reaction mixture was incubated at 39°C for 40 min. The RT-RPA reaction mixture comprised 29.5 μL rehydration buffer, 2.4 μL forward and reverse primers (10 μM each), 1 μL SuperScript IV Reverse Transcriptase (Invitrogen, Waltham, Massachusetts, USA), 1 μL RNase inhibitor

**Table 1. RT-RPA primer and gRNA sequences.**

|  |  | Target gene | Sequence | PAM |
|---|---|---|---|---|
| **RT-RPA primer** | OT1-F | 16S rRNA | GGCTTAACCCTGGAACTGCTTCTAAAACTG |  |
|  | OT1-R | 16S rRNA | CTTTCGCCACTGGTGTTCCTTCTAATATCT |  |
|  | OT2-F | 16S rRNA | GTGCTAGATATTGGGGGATTTTTCTTTCAG |  |
|  | OT2-R | 16S rRNA | TTGGTAAGGTTTTTCGCGGATCATCGAATT |  |
| **gRNA** | OT1 | 16S rRNA | TAGTGTAGAGGTAAAATTCT | TTTC |
|  | OT2 | 16S rRNA | GTAGCTAACGCATTAAGCAC | TTTC |

(Enzynomics, Daejeon, Korea), and 2.5 μL of 280 mM magnesium acetate, with samples and nuclease-free water added to obtain a final volume of 50 μL. Each RT-RPA reaction mixture was incubated at 42°C for 40 min.

LbCas12a trans-cleavage assays were performed as previously described [24,27]. To design gRNA for detection the nine types of *O. tsutsugamushi*, the previously reported *O. tsutsugamushi* 16S rRNA sequences were obtained from NCBI and screened for shared subsequences. From this analysis, we designed and synthesized two gRNAs, OT1 and OT2 which corresponded to nucleotides 615–634 and 791–810, respectively (Bioneer, Daejeon, Korea) and were expected to specifically detect the *O. tsutsugamushi* 16S rRNA of all nine genotypes but not those of other bacterial genomes (Table 1). LbCas12a (New England Biolabs, Ipswich, Massachusetts, USA) and LbCas12a-specific gRNA were mixed with 1× NEBuffer 2.1 to final concentrations of 50 nM and 62.5 nM, respectively, and then mixed together and incubated at 37°C for 30 min to generate LbCas12a-gRNA complexes. For fluorescence assays, 2 μL of RT-RPA products, 80 μL of 1× NEBuffer 2.1, 18 μL of LbCas12a-gRNA complex and 2 μL of 10 μM FQ-labeled reporter (/56-FAM/TTATT/3IABkFQ/; Integrated DNA Technologies, Coralville, IA, USA) were dispensed directly to 96-well microplates, which were incubated at 37°C for 20 min. Fluorescence measurements were taken at the beginning and end of the incubation period using the Glomax Discover Microplate Reader (Promega, Madison, WI, USA) equipped with version 4.0.0 software (λex, 485 nm; λem, 535 nm). To simplify the visualization of results, we used a lateral flow assay (LFA). For the LFA, 2 μL of RT-RPA products, 40 μL of 1× NEBuffer 2.1, 36 μL of LbCas12a-gRNA complex and 2 μL of 10 μM lateral flow cleavage reporter (/56-FAM/TTATT/3Bio/; Integrated DNA Technologies) were combined in an Eppendorf tube and incubated at 37°C for 20 min. A Milenia HybriDetect 1 lateral flow strip (Milenia Biotec, Giesesen, Germany) was applied to the incubated sample according to the manufacturer's instructions. The results were examined after 2 min and interpreted as previously described [27].

## One-pot OT DETECTR

To prevent contamination and increase the field-applicability of OT DETECTR, we designed a one-pot OT DETECTR that is performed in a single tube. For this method, the LbCas12a-gRNA complex was generated by combining 3 μL of 1μM LbCas12a, 1 μL of 10 μM gRNA, 2 μL of 10 μM FQ-labeled reporter, and 54 μL of 1× NEBuffer 2.1 in an Eppendorf tube, which was then fitted with a 0.22 μm filter tube (SPL Life Sciences, Pocheon, Korea). The above-described RT-RPA mixture was loaded to the 0.22 μm-filter tube, and the assemblage was incubated at 42°C for 40 min. The LbCas12a-gRNA complex and RT-RPA product were then mixed via centrifugation (13,000 rpm, 1 min) and further incubated at 42°C for 20 min (Fig 1B). Fluorescence results were obtained at the start and end of incubation (λex, 485 nm; λem, 535 nm).

## Bacteria

Bacterial cultures of *Staphylococcus aureus* (SA), *Klebsiella pneumoniae* (KP), *Salmonella enteritidis* (SE), and *E. coli* were grown overnight at 37°C in 5 mL LB broth, and bacterial RNA was extracted using a bacterial RNA extraction kit (Bioneer, Daejeon, Korea). SA, KP and SE were kindly provided by Dr. Young-Seo Park (Gachon University, Korea).

*O. tsutsugamushi* (Ikeda strain; National Culture Collection for Pathogens, NCCP) was inoculated to a monolayer of L929 cells. At 4 days post-infection, the cells were collected and homogenized with sterile glass beads using a TissueLyser II (Qiagen, Hilden, Germany). The homogenate was centrifuged at 500×g for 5 min, the supernatant was collected, and cell-free

**Table 2. PCR primers used for *in vitro* transcription.**

| Bacteria | Primer | Sequence |
|---|---|---|
| *O. tsutsugamushi* | T7 OT-F' | TAATACGACTCACTATAGGGAGAAACGAACGCTGGCGG |
|  | T7 OT-R' | GCAGGTTCCCCTACGGCTACCTTGTTACGACTTTAC |
| *Rickettsia* spp. | T7 RS-F' | TAATACGACTCACTATAGGGAGAGGCTTAACCTCGGAA |
|  | T7 RS-R' | CCTTCGCCACCGGTGTTCCTCCTAATATCTAAGAATTT |

bacteria were collected by high-speed centrifugation at 6000×g for 30 min and resuspended in DMEM. The infectivity titer of the inoculum was determined using an immunofluorescence assay, and the infected cell-counting unit (ICU) was calculated as follows: ICU = (total number of cells used for infection) × (ratio of infected cells to counted cells) × (dilution fold of *O. tsutsugamushi* Ikeda inoculum) [32]. The *O. tsutsugamushi* genomic DNA and RNA were extracted using Qiagen DNeasy Blood & Tissue Kit and Qiagen RNeasy Mini Kit (Qiagen, Hilden, Germany), respectively. The *O. tsutsugamushi* genomic DNA and RNA were used for comparative testing of *O. tsutsugamushi* DNA molecules and RNA molecules content rates.

### *In vitro* transcription of *O. tsutsugamushi* and *Rickettsia* spp. 16S rRNAs

The 16S rRNA was synthesized by an *in vitro* transcription (IVT) reaction and used to test the limit of detection (LoD) and specificity for each version of OT DETECTR. The 16S rRNA sequences of 8 strains *O. tsutsugamushi*: Karp, Kuroki, TA763, Gilliam, Kawasaki, Japanese Gilliam, Kato, and Shimokoshi (GenBank accession numbers: NR_025860.1, D38626.1, NZ_OUNA00000000.1, D38622.1, D38625.1, AP008981.1, NZ_LS398550.1, D38627.1, respectively; NCBI), were synthesized in plasmid form (Macrogen, Seoul, Korea) and PCR amplified using primers with T7 promoter sequences (Table 2). The primers and gRNA designed for *O. tsutsugamushi* detection showed similarity to regions in the 16S rRNA of *Rickettsia* spp., including species from the spotted fever and typhus groups (S1 Table). This region was identical across all 18 *Rickettsia* species analyzed, and the sequence was synthesized as plasmid DNA for experimental use. The 16S rRNA sequence was then amplified using primers with T7 promoter sequences (Table 2). For the IVT reaction, the mMESSAGE mMACHINE T7 Transcription kit (Invitrogen, Waltham, MA, USA) was applied according to the manufacturer's instructions, with the above-described PCR products used as the template. The synthesized 16S rRNA was purified with a GeneJET RNA Cleanup and Concentration Micro Kit (Thermo Scientific, Waltham, MA, USA).

### Preparation of *O. tsutsugamushi* clinical samples

All *O. tsutsugamushi* cases were confirmed using nested PCR, a method commonly employed for its diagnosis in Korea, at Jeju National University Hospital (Jeju, Korea). To confirm *O. tsutsugamushi* infection in clinical samples, bacterial DNA and RNA were extracted from whole blood using a QIAamp DNA Blood Mini kit (Qiagen, Hilden, Germany) and QIAamp RNA Blood Mini kit (Qiagen, Hilden, Germany), respectively. Nested PCR was performed as previously described [11] to detect the gene encoding the TSA56 of *O. tsutsugamushi* in extracted bacterial DNA, and OT DETECTR was performed to detect the 16S rRNA of *O. tsutsugamushi* in bacterial RNA.

### Statistical analysis

Data are expressed as mean ± standard deviation (SD) of three independent experiments. The significance of differences between two means was determined with the Student's t-test using

GraphPad Prism 7 (GraphPad Software, Inc., Sandiego, CA. USA). *P*-values were determined by unpaired two-sided Student's t-tests. *P*-values < 0.05 were considered statistically significant.

## Results

### Designing a CRISPR-Cas12a-based assay for *O. tsutsugamushi*

We herein designed a CRISPR-Cas12a-based diagnostic assay for rapid and accurate diagnosis of *O. tsutsugamushi* 16S rRNA and designated it OT DETECTR. The 16S rRNA of *O. tsutsugamushi* was selected as the target because it has a stable secondary structure and exists in more copies than a DNA target would. We screened previously reported sequences of *O. tsutsugamushi* to design RPA primer-gRNA sets that recognized the 16S rRNAs of all *O. tsutsugamushi* genotypes but not those of other organisms, and selected two sets, OT1 and OT2 (S1 Fig.). The *O. tsutsugamushi* 16S rRNA was extracted using a column-based extraction kit, amplified by isothermal amplification (RT-RPA), and detected by leveraging Cas12a trans-cleavage activity (Fig 1A), which was determined by either fluorescence assay or LFA. To prevent contamination and increase the field applicability of our method, we further developed a one-pot OT DETECTR in which all reactions are performed in a single tube at the same temperature (Fig 1B), as compared to the two-pot OT DETECTR.

Using nucleic acids extracted from *O. tsutsugamushi*, we found that our 16S rRNA-based diagnostic method was more sensitive for RNA than DNA (Fig 2). DNA and RNA were extracted from various infected-cell counting units (ICUs) of *O. tsutsugamushi*, and the sensitivity of OT DETECTR was compared using a detection versus non-detection approach. The OT1 set could detect *O. tsutsugamushi* DNA at a titer of $10^4$ ICU per reaction and RNA at a titer of $10^3$ ICU per reaction, indicating that RNA target detection was 10 times more sensitive than DNA target detection (Fig 2A). The OT2 set detected both DNA and RNA of *O. tsutsugamushi* at a titer of $10^4$ ICU per reaction and thus showed the same sensitivity (Fig 2B). Based on these results, we used the OT1 primer-gRNA set in subsequent experiments.

### Measuring of the limit of detection (LoD) for OT DETECTR

To determine the LoD of the assay, two-pot and one-pot OT DETECTR methods combined with a fluorescence assay were performed using varying copy numbers of *in vitro* transcribed *O. tsutsugamushi* 16S rRNA from the Karp strain. Both OT DETECTRs could detect as little as 100 RNA copies per reaction (Fig 3A and 3B). The one-pot OT DETECTR showed higher fluorescence values compared to the two-pot DETECTR: The fluorescence values of the one-pot OT DETECTR were 70% and 187% higher than those of the two-pot DETECTR for $10^3$ and $10^2$ RNA copies per reaction, respectively.

### Specificity analysis of OT DETECTR

To determine the specificity of OT DETECTR, it was tested on genomic RNA from *O. tsutsugamushi*, *S. aureus*, *K. pneumoniae*, *S. enteritidis*, and *E. coli*, as well as on *in vitro* transcribed 16S rRNA from *Rickettsia* spp. (Fig 4). The two-pot and one-pot OT DETECTRs combined with a fluorescence assay both yielded positive results with RNA of *O. tsutsugamushi*, but not with that of *S. aureus*, *K. pneumoniae*, *S. enteritidis*, *E. coli* or *Rickettsia* spp.. Thus, OT DETECTR specifically detected *O. tsutsugamushi* without cross-reacting with *S. aureus*, *K. pneumoniae*, *S. enteritidis*, *E. coli* or *Rickettsia* spp.. The OT DETECTR's ability to detect various *O. tsutsugamushi* strains was further evaluated. Both two-pot and one-pot OT DETECTR

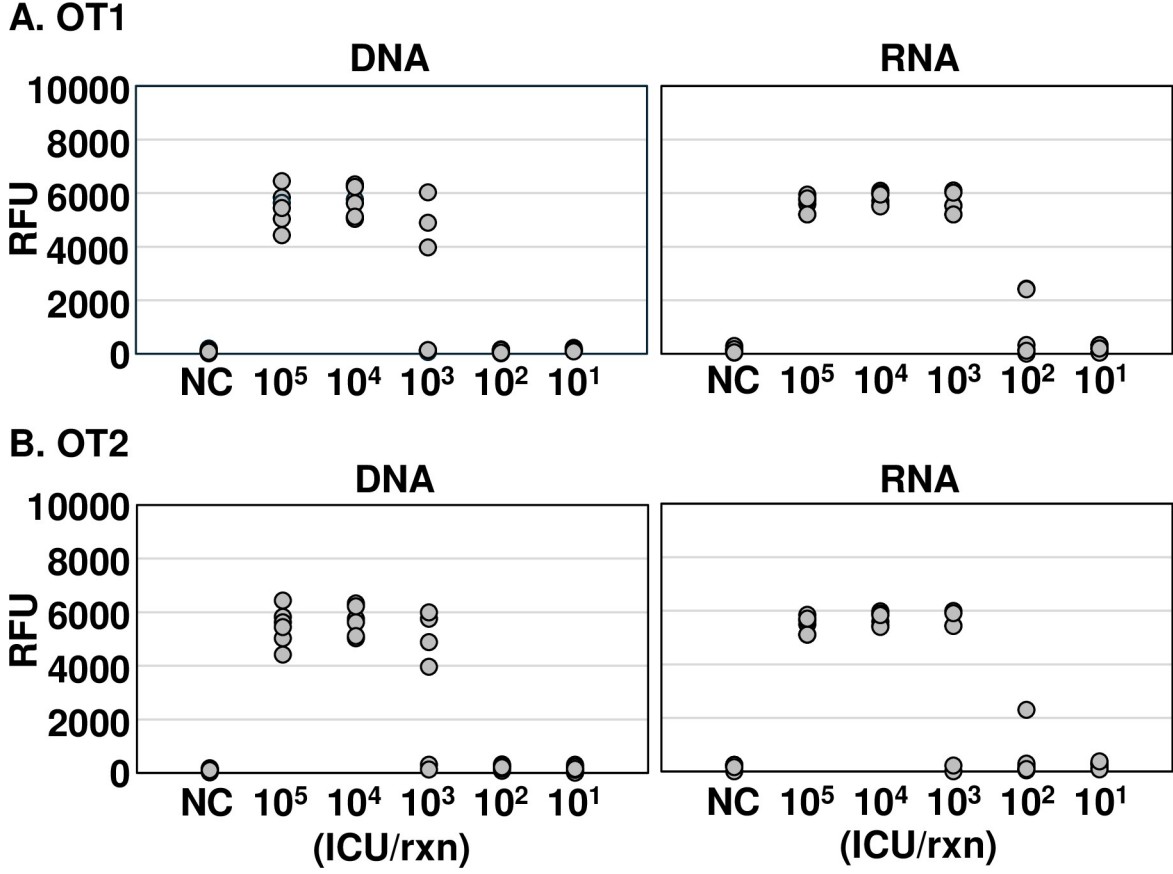

**Fig 2. Comparative analysis of the sensitivity of the OT DETECTRs to *O. tsutsugamushi* DNA and RNA.** Various ICUs ($10^5$ to $10^1$) of *O. tsutsugamushi* were lysed with a column-based extraction kit, and nucleic acids (DNA and RNA) were amplified via RT-RPA with primer sets specific for the 16S rRNA gene. RT-RPA products were detected using two-pot OT DETECTR combined with a fluorescence assay. Shown are results obtained using gRNAs (A) OT1 and (B) OT2 (*n* = 6 replicates). RFU, relative fluorescence unit; NC, *no template control; ICU, infected cell-counting unit.*

successfully detected major strains, including Karp, Kuroki, TA763, Gilliam, Kawasaki, Japanese Gilliam, Kato, and Shimokoshi (S2 Fig).

### Applicability of OT DETECTR to clinical samples

To evaluate the clinical applicability of the one-pot and two-pot OT DETECTRs, we compared the results obtained from samples of patients confirmed to be infected with *O. tsutsugamushi* by nested PCR, a commonly used confirmatory method for diagnosing scrub typhus in Korea. A total of 125 clinical samples were evaluated for *O. tsutsugamushi* infection via nested PCR (S3 Fig). Among them, 50 clinical samples were confirmed to be positive for *O. tsutsugamushi* infection and 75 were confirmed to be negative (S3 Fig). The two-pot DETECTR combined with fluorescence assay or LFA detected 49 of 50 *O. tsutsugamushi* infection-positive samples (the long exception was patient #27) (Fig 5A). Similarly, the one-pot DETECTR combined with a fluorescence assay also confirmed infection in all clinical samples positive for *O. tsutsugamushi* infection except for patient #27 (Fig 6A). Among clinical samples negative for *O. tsutsugamushi*, the two-pot and one-pot DETECTRs yielded the same results as the nested PCR (Figs 5B and 6B). The application of OT DETECTR to clinical samples and its concordance analysis with nested PCR are summarized in Tables 3 and 4, respectively. Patient demographic

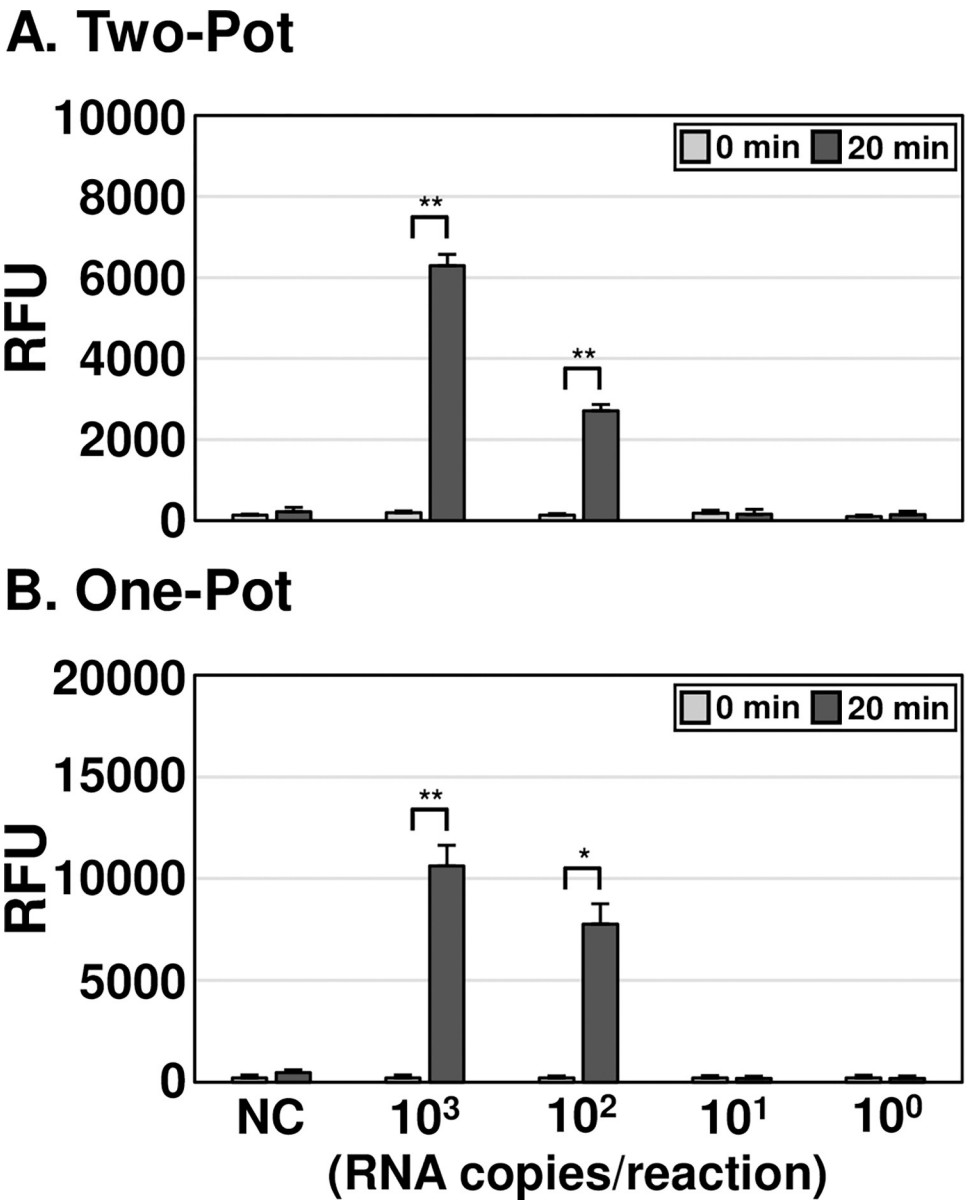

**Fig 3. LoD analyses of OT DETECTRs.** (A, B) To determine the LoD of each OT DETECTR, different copy numbers ($10^3$ to $10^0$) of IVT *O. tsutsugamushi* 16S rRNA were used as templates for the (A) two-pot and (B) one-pot OT DETECTR combined with a fluorescence assay. Values are presented as means ± s.d. (error bars) ($n$ = 3 replicates; ** $P < 0.01$, * $P < 0.05$ between samples, two-sample *t*-test). RFU, relative fluorescence unit; NC, *no template control*.

and clinical characteristics, including age, gender, immunosuppression status, Charlson Comorbidity Index (CCI), and days since illness onset, are detailed in S2 Table.

## Discussion

In this study, we developed a CRISPR-Cas12a-based diagnostic method for *O. tsutsugamushi* (OT DETECTR). We targeted the 16S rRNA, which is present in more copies than DNA, offers a stable secondary, and was found to yield a detection sensitivity more than 10 times greater than that for DNA when targeted by OT DETECTR. This enables reliable detection

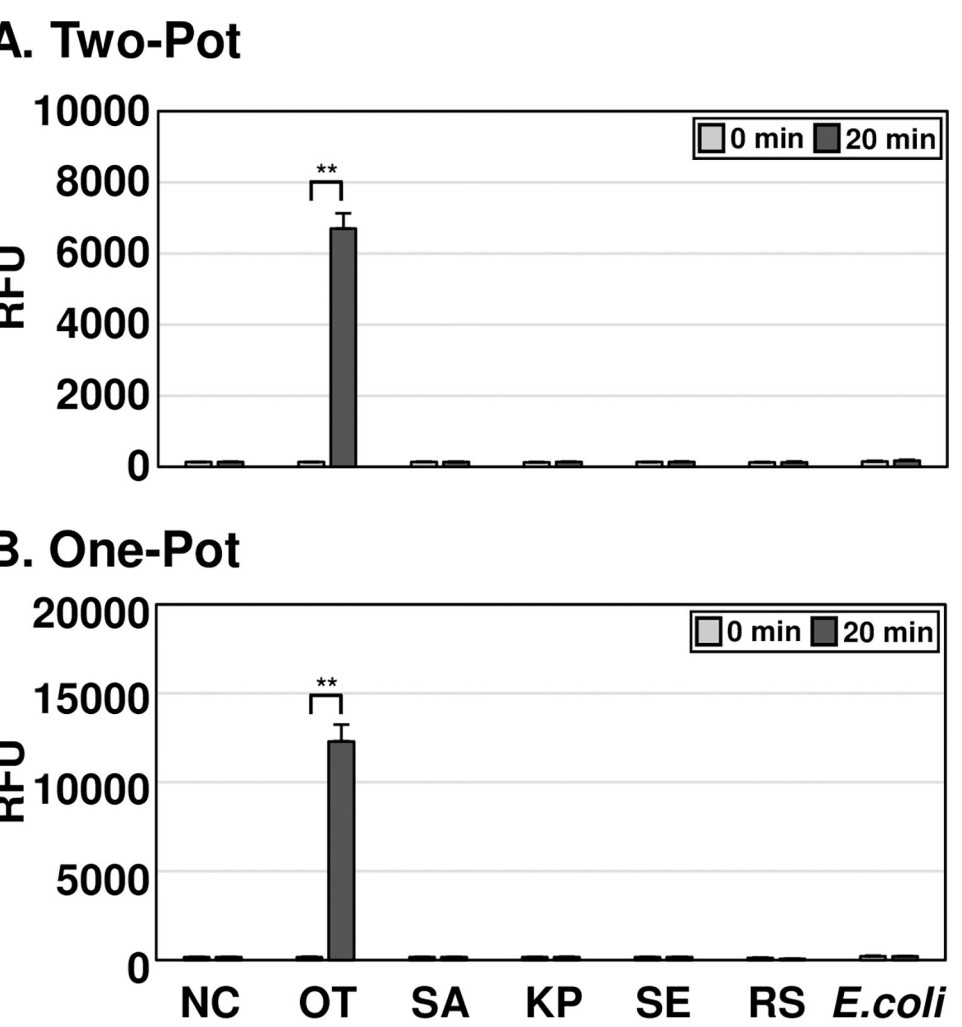

**Fig 4. Specificity analyses of OT DETECTRs.** To determine the specificities of OT DETECTRs, 100 ng of RNA from each bacterial species and $10^5$ copies of RNA from *Rickettsia* spp. were amplified using RT-RPA. The Cas12a trans-cleavage activity assay was subsequently performed, and the results were analyzed through fluorescence measurements. Values are presented as means ± s.d. (error bars) (*n* = 3 replicates; ** *P* < 0.01, between samples, two-sample *t*-test). RFU, relative fluorescence unit; NC, *no template control; SA, Staphylococcus aureus*; KP, *Klebsiella pneumoniae*; SE, *Salmonella enteritidis*; RS, *Rickettsia* spp.; OT, *Orientia tsutsugamushi*.

even in environments with low pathogen concentrations. Our method detected all *O. tsutsugamushi* genotypes with known sequences and exhibited high specificity (100%). The analytical sensitivity of this diagnostic method was $10^2$ RNA copies per reaction, which is approximately 10-fold lower than that of the nested PCR used in conventional diagnostic methods [33]. Although nested PCR amplifies trace amounts of target DNA using two rounds of PCR, it has the disadvantages of requiring additional reaction time and posing a higher risk of contamination. Compared to the long reaction time of 3 hours 45 minutes for nested-PCR, OT DETECTR utilizes isothermal amplification and completes within 1 hour, making it a more suitable diagnostic technology for scrub typhus, where rapid diagnosis is critical.

Meanwhile, in an analysis conducted with blood samples from patients, the OT DETECTR exhibited a clinical sensitivity of 98% (49/50), which was highly consistent with the sensitivity of the nested PCR analysis (98%). The single false-negative sample (positive clinical sample no. 27) had a low purity ratio of absorbance at 260 nm and 280 nm (0.5), compared to other

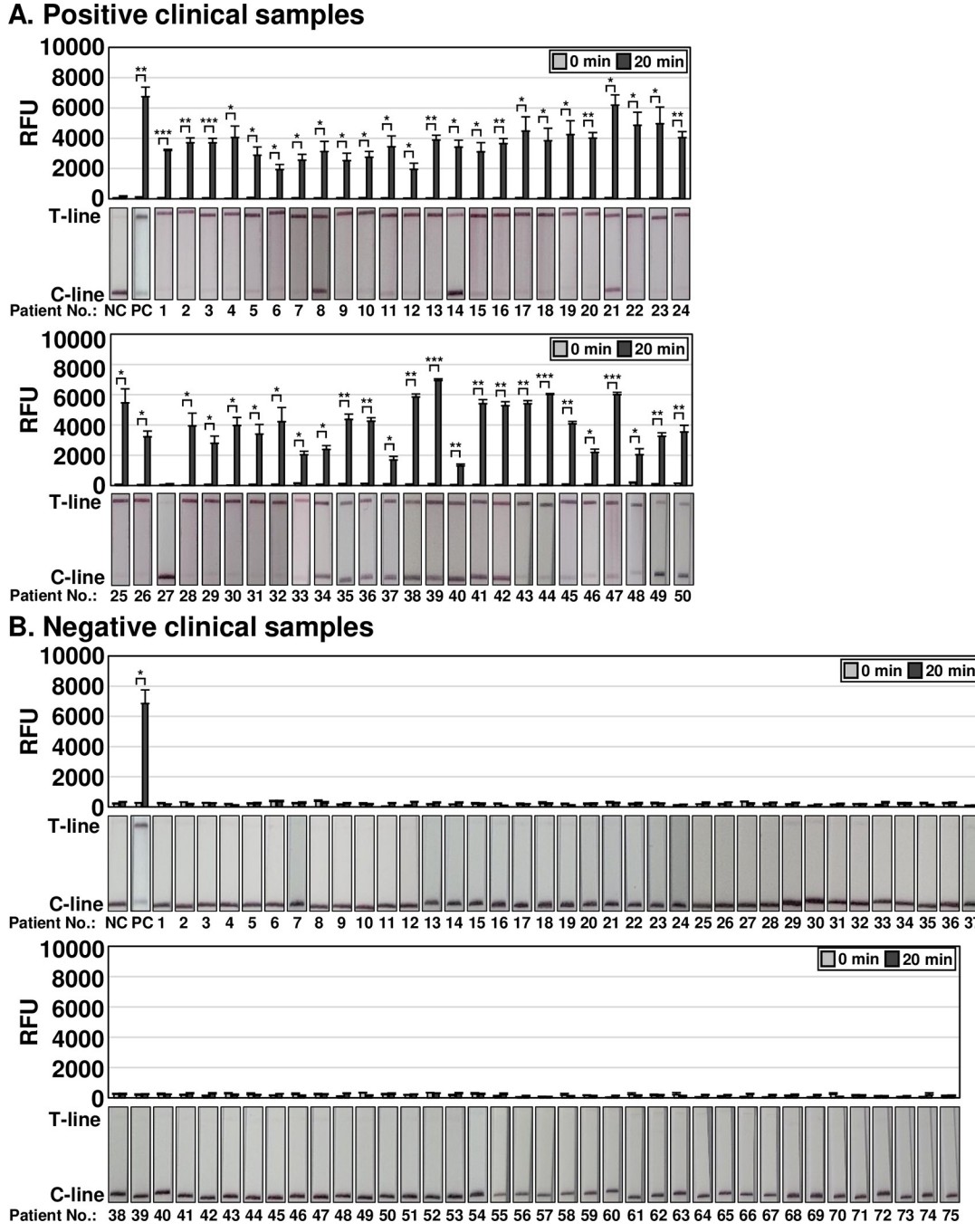

**Fig 5. Clinical applicability analysis of two-pot OT DETECTR.** Two-pot OT DETECTR was performed on clinical samples from (A) 50 patients confirmed positive or (B) 75 patients confirmed negative for *O. tsutsugamushi* infection by nested PCR. RNA was extracted from whole blood, RT-RPA was performed, the Cas12a trans-cleavage activity assay was performed, and the results were determined by a fluorescence assay or LFA. Genomic RNA of *O. tsutsugamushi* (100 ng) was used as a positive control. Values are presented as means ± s.d. (error bars) (*n* = 3 replicates; *** *P* < 0.001, ** *P* < 0.01, * *P* < 0.05 between samples, two-sample *t*-test). RFU, relative fluorescence unit; C-line, control line; T-line, test line; NC, *no template control; PC, positive control*.

samples with ratios of 1.6 or higher. This suggests that the assay's performance may be affected by sample quality. Moreover, while nested PCR requires a thermocycler and a reaction time of 3 h 45 min, OT DETECTR can provide results within 1 hour and uses an isothermal reaction.

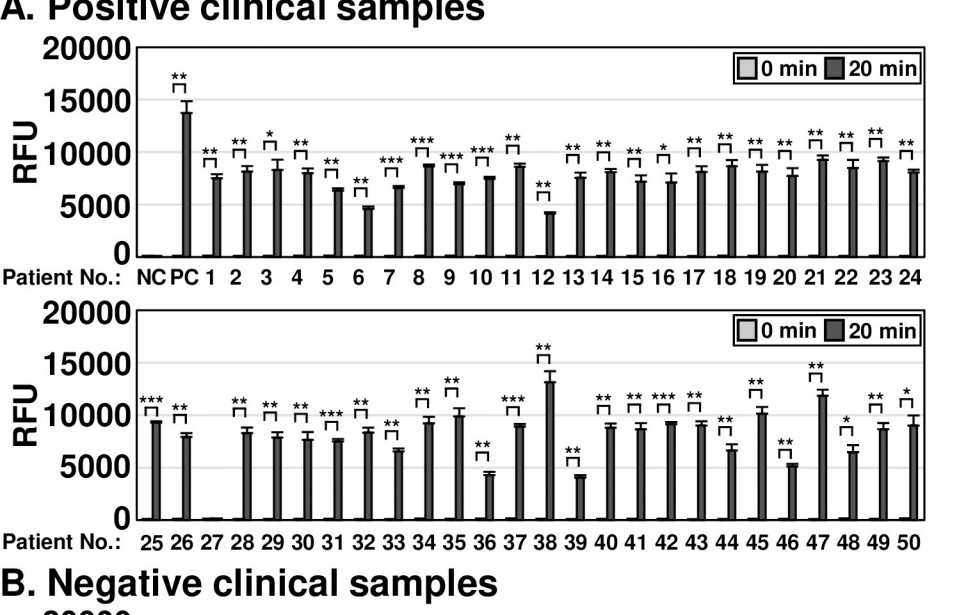

## A. Positive clinical samples

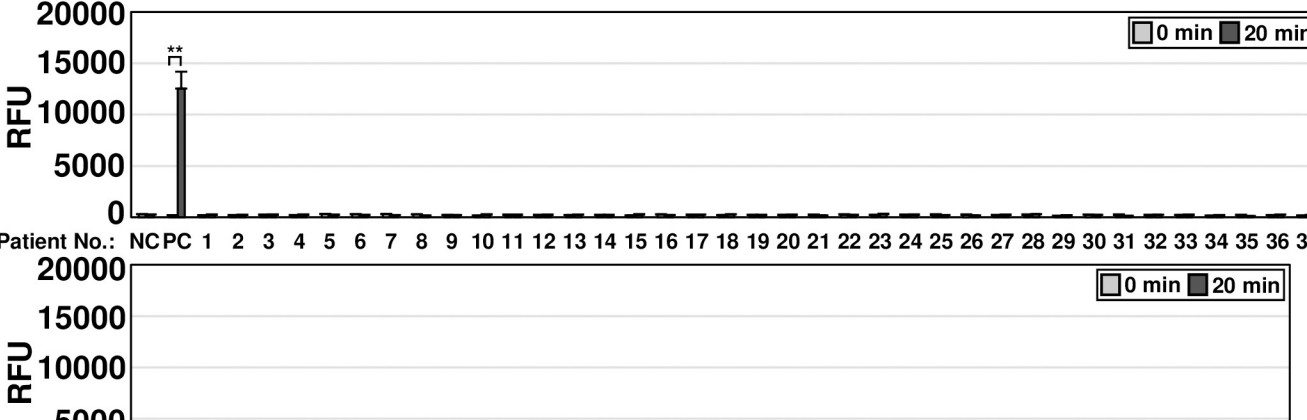

## B. Negative clinical samples

**Fig 6. Clinical applicability analysis of one-pot OT DETECTR.** One-pot OT DETECTR combined with a fluorescence assay was performed on clinical samples from (A) 50 patients confirmed positive or (B) 75 patients confirmed negative for *O. tsutsugamushi* infection by nested PCR. Genomic RNA of *O. tsutsugamushi* (100 ng) was used as a positive control. Values are presented as means ± s.d. (error bars) ($n = 3$ replicates; *** $P < 0.001$, ** $P < 0.01$, * $P < 0.05$ between samples, two-sample *t*-test). RFU, relative fluorescence unit; NC, *no template control; PC, positive control.*

The ability of OT DETECTR to diagnose *O. tsutsugamushi* infection with high sensitivity and specificity in such a short time without the need for specialized equipment supports its applicability for the rapid diagnosis of scrub typhus to facilitate timely and appropriate treatment.

**Table 3. Performance of OT DETECTR on clinical samples.**

|  | True positive | False negative | True negative | False positive | Sensitivity (95% CI) | Specificity (95% CI) |
|---|---|---|---|---|---|---|
| Two-pot OT DETECTR | 49 | 1 | 75 | 0 | 98% (89.35%~99.5%) | 100% (95.2%~100%) |
| One-pot OT DETECTR | 49 | 1 | 75 | 0 | 98% (89.35%~99.5%) | 100% (95.2%~100%) |

**Table 4. Concordance analysis between nested PCR and OT DETECTR for *O. tsutsugamushi* detection.**

| Detection Outcome | Nested PCR positive | Nested PCR negative | Total | Percentage (%) |
|---|---|---|---|---|
| Two-pot OT DETECTR positive | 49 | 0 | 49 | *PPA: 98% |
| Two-pot OT DETECTR negative | 1 | 75 | 76 | **NPA: 100% |
| Total | 50 | 75 | 125 | |
| One-pot OT DETECTR positive | 49 | 0 | 49 | *PPA: 98% |
| One-pot OT DETECTR negative | 1 | 75 | 76 | **NPA: 100% |
| Total | 50 | 75 | 125 | |

*PPA: positive percent agreement

**NPA: Negative percent agreement

Beyond comparisons with nested PCR, evaluating OT DETECTR against quantitative methods like qPCR could further enhance its clinical applicability. While qPCR provides quantitative data, such as bacterial load in copies/ml, OT DETECTR focuses on rapid, qualitative detection, offering distinct advantages in resource-limited settings. A direct comparison with qPCR could provide deeper insights into OT DETECTR's analytical performance and clinical applicability, especially in contexts requiring quantitative measurements. Future studies will include such comparisons to comprehensively assess OT DETECTR's strengths and limitations.

In the conventional DETECTR protocol, the isothermal nucleic acid amplification and trans-cleavage reaction are conducted in separate tubes at different temperatures. This opens the possibility of sample contamination and/or error during the transfer of reaction materials. To address these issues, we devised a one-pot OT DETECTR. We placed the trans-cleavage reaction mixture at the bottom of an Eppendorf tube, inserted a filter into the tube, and loaded the nucleic acid amplification reaction mixture (i.e., RPA reaction mixture) into the filter. At a single temperature, the amplification step took place on the filter, a brief centrifugation was used to move the amplified product to the bottom of tube, and then the trans-cleavage reaction took place at the bottom of the tube. The devised one-pot DETECTR showed no difference in specificity and sensitivity compared to the conventional two-pot DETECTR, and actually exhibited higher fluorescence values in measuring trans-cleavage activity. This suggests that a one-pot system could perform at least as well as the corresponding two-pot system while potentially preventing contamination and/or errors during the diagnostic process and thereby resolving issues of false negatives. Currently, Inogenix Inc. is evaluating and developing an OT DETECTR-based *O. tsutsugamushi* diagnostic kit that offers these features. Further studies are warranted to use the diagnostic kit for additional clinical samples and to evaluate its performance in field-based testing.

For the OT DETECTR presented in this study to be used for point-of-care testing (POCT), several technological barriers need to be overcome. First, it must be easy to extract nucleic acids from the blood of suspected patients. To tackle the challenge of nucleic acid extraction in resource-limited settings, we plan to investigate simplified methods, such as heat-based extraction techniques. These approaches aim to streamline sample preparation and improve the test's practicality in field conditions. Second, it should use a reaction temperature setting that can be achieved using a battery-based device or an air-activated hand warmer. Third, it should involve the sequential progression of isothermal nucleic acid amplification and trans-cleavage reaction using microfluidics or a nanoplatform. Finally, it should use LFA or a mobile device-based technique for measuring fluorescence. If these technologies are developed and integrated, it is expected that a CRISPR-Cas12a-based POCT can be developed for use in low- and

middle-income countries (LMICs), as well as in rural and remote areas with fragile healthcare systems.

Recently, Bhardwaj *et al.* reported a conventional CRISPR-Cas12a-based detection method, combined with LFA, which could detect a single gene copy of the genomic DNA from *O. tsutsugamushi* Karp and Gilliam strains [34]. Since LFA has the drawback of displaying a positive band depending on reaction time [35], further studies are required to verify these findings using quantitative methods, such as fluorescence assays.

Several studies have reported that a CRISPR-Cas13a-based assay can detect RNA without nucleic acid amplification [36, 37]. Therefore, we also tested a CRISPR-Cas13a-based assay with OT1 gRNA on a subset of clinical samples confirmed positive for *O. tsutsugamushi* infection. However, we failed to obtain any results (S4 Fig). We further used the two gRNAs (OT1 and OT2) for a CRISPR-Cas13a-based assay because the sensitivity of the assay was reported to be enhanced by the use of multiple primers [37]. However, despite many attempts, we were unable to replicate the previously reported success with Cas13a (S4 Fig).

## Supporting information

**S1 Fig.** Sequence alignment of the (A) RPA primers and (B) gRNAs used in this study, relative to the different genotypes of *O. tsutsugamushi*. Sequences of OT1-R and OT2-R are shown in the reverse-complement orientation.
(TIF)

**S2 Fig. Detection of major *O. tsutsugamushi* strains.** The *in vitro* transcribed RNA fragments from the 16S rRNA sequences of major *O. tsutsugamushi* strains (Karp, Kuroki, TA763, Gilliam, Kawasaki, Japanese Gilliam, Kato, and Shimokoshi) were used to evaluate (A) Two-pot DETECTR and (B) One-pot DETECTR. Values are presented as means ± s.d. (error bars) ($n = 3$ replicates; *** $P < 0.001$, ** $P < 0.01$, * $P < 0.05$ between samples, two-sample *t*-test). RFU, relative fluorescence unit; NC, no template control.
(TIF)

**S3 Fig. Nested PCR-based analysis of *O. tsutsugamushi* infection in clinical samples.** Nested PCR was performed to assess *O. tsutsugamushi* infection in 125 clinical samples from patients believed to be positive or negative for infection, and the product (483 bp) was visualized via gel electrophoresis. Of the total 125 samples, (A) 50 samples were confirmed positive for *O. tsutsugamushi* infection, and (B) 75 samples were confirmed negative for *O. tsutsugamushi* infection.
(TIF)

**S4 Fig. Evaluating the applicability of LwCas13a-based detection using clinical samples from patients with confirmed *O. tsutsugamushi* infection.** Clinical samples from 10 patients with confirmed *O. tsutsugamushi* infection were analyzed using OT DETECTR with (A) Cas13a-OT1 gRNA or (B) multiple gRNAs (Cas13a-OT1 and Cas13a-OT2). Values are presented as means ± s.d. (error bars) ($n = 3$ replicates; * $p < 0.05$ between samples, two-sample *t*-test). RFU, relative fluorescence unit; NC, no template control; PC, Positive control (*in vitro* transcribed *O. tsutsugamushi* 16S rRNA)
(TIF)

**S1 Table. 16S rRNAs of *Rickettsia* spp.**
(DOCX)

**S2 Table. Information on patients with positive *O. tsutsugamushi* infection.**
(DOCX)

**S1 Materials and Methods. Cas13a cleavage reaction.**
(DOCX)

## Author Contributions

**Conceptualization:** Bum Ju Park, Yoon-Jae Song.

**Data curation:** Bum Ju Park.

**Formal analysis:** Bum Ju Park, Eun Jin Bae, Su Yeon Kang, Sunghoon Park.

**Funding acquisition:** Yoon-Jae Song.

**Investigation:** Bum Ju Park, Eun Jin Bae, Su Yeon Kang, Sunghoon Park, Kyeo Re Han, Jae Myun Lee, Hyeyoung Lee, Yoon-Jae Song.

**Methodology:** Bum Ju Park, Eun Jin Bae, Su Yeon Kang, Sunghoon Park, Keun Hwa Lee, Jae Myun Lee, Hyeyoung Lee, Yoon-Jae Song.

**Project administration:** Yoon-Jae Song.

**Resources:** Sang Taek Heo, Misun Kim, Jeong Rae Yoo, Keun Hwa Lee.

**Software:** Bum Ju Park.

**Supervision:** Keun Hwa Lee, Jae Myun Lee, Hyeyoung Lee, Yoon-Jae Song.

**Validation:** Bum Ju Park.

**Visualization:** Bum Ju Park.

**Writing – original draft:** Bum Ju Park, Keun Hwa Lee, Jae Myun Lee, Yoon-Jae Song.

**Writing – review & editing:** Bum Ju Park, Yoon-Jae Song.

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
