## [Decision Letter · Decision Letter 0]

24 Oct 2024

PNTD-D-24-01350A CRISPR-Cas12a-based universal rapid scrub typhus diagnostic method targeting 16s rRNA of Orientia tsutsugamushiPLOS Neglected Tropical Diseases Dear Dr. Song, Thank you for submitting your manuscript to PLOS Neglected Tropical Diseases. After careful consideration, we feel that it has merit but does not fully meet PLOS Neglected Tropical Diseases's publication criteria as it currently stands. Therefore, we invite you to submit a revised version of the manuscript that addresses the points raised during the review process. Please submit your revised manuscript within 60 days Dec 23 2024 11:59PM. If you will need more time than this to complete your revisions, please reply to this message or contact the journal office at plosntds@plos.org. Please include the following items when submitting your revised manuscript:*
A rebuttal letter that responds to each point raised by the editor and reviewer(s). You should upload this letter as a separate file labeled 'Response to Reviewers'. This file does not need to include responses to any formatting updates and technical items listed in the 'Journal Requirements' section below.*
A marked-up copy of your manuscript that highlights changes made to the original version. You should upload this as a separate file labeled 'Revised Manuscript with Track Changes'.*
An unmarked version of your revised paper without tracked changes. You should upload this as a separate file labeled 'Manuscript'. If you would like to make changes to your financial disclosure, competing interests statement, or data availability statement, please make these updates within the submission form at the time of resubmission. Guidelines for resubmitting your figure files are available below the reviewer comments at the end of this letter. We look forward to receiving your revised manuscript. Kind regards, Yong QiAcademic EditorPLOS Neglected Tropical Diseases Georgios PappasSection EditorPLOS Neglected Tropical Diseases

Shaden Kamhawi

co-Editor-in-Chief

Paul Brindley

co-Editor-in-Chief

 **Journal Requirements:** **Additional Editor Comments (if provided):****Reviewers' Comments:** Reviewer's Responses to Questions

**Key Review Criteria Required for Acceptance?**

**Methods**

-Are the objectives of the study clearly articulated with a clear testable hypothesis stated?

-Is the study design appropriate to address the stated objectives?

-Is the population clearly described and appropriate for the hypothesis being tested?

-Is the sample size sufficient to ensure adequate power to address the hypothesis being tested?

-Were correct statistical analysis used to support conclusions?

-Are there concerns about ethical or regulatory requirements being met?

Reviewer #1: While the objectives are generally clear, the study design and population description have limitations, particularly in terms of sample size and generalizability. The statistical analyses, though appropriate, could be more robustly presented, and no significant ethical concerns are evident.

The methods section of the manuscript has several deficiencies that could affect the robustness and generalizability of the results. First, the sample size of 125 clinical samples, while providing an initial proof of concept, is too small to offer strong statistical power or broad applicability across different populations and regions. The authors should consider increasing the sample size and including a more diverse set of samples, both geographically and in terms of disease severity, to better evaluate the test’s performance. Issues such as age, sex, days of illness, incredibly important in diagnostic circumstances and this information should be presented in the manuscript.

Additionally, the lack of field-based testing limits the study’s ability to address its objective of developing a point-of-care diagnostic tool. Testing under real-world conditions in resource-limited settings would provide critical insights into the feasibility and robustness of the test in practical applications.

Moreover, the statistical analysis, while appropriate in the use of Student’s t-test, lacks detailed presentation of critical values such as confidence intervals and p-values. The authors should enhance their statistical analysis by including these details and ensuring that significance is clearly marked in their results, especially within figures and tables. This would strengthen the reliability of the findings and allow for a clearer interpretation of the data.

Reviewer #2: (No Response)

Reviewer #3: The methods section is well-written, and the objectives are clear and appropriate. It would be helpful to specify the instrumentation and software used for fluorescence measurement in the assay. Additionally, could you elaborate on the rationale behind selecting specific strains like S. aureus and K. pneumoniae for specificity testing? Why were other Rickettsia species or other vector-borne pathogens not included?

Reviewer #4: 1.The term "universal diagnosis" should be validated with additional strains of Orientia spp. to ensure broader applicability. or it should demonstrate a board range of detection across known O. tsutsugamushi genotypes in the manuscript.

2.In the study design you mentioned “We designed primer-gRNA sets to detect nine genotypes of O. tsutsugamushi”

-Have you validated the detection efficacy for all nine genotypes in real samples?

3.For test validation, it would be beneficial to explain why Staphylococcus aureus (SA), Klebsiella pneumoniae (KP), Salmonella enteritidis (SE), and E. coli were used. Additionally, consider including more closely related bacteria from the Rickettsia group to strengthen the validation. If not please address why they were not include?

4.Do you plan to include a positive control or quality control for the each test runs? or provide explanation how will you validate each run.

5.Consider conducting a concordance analysis for positive predictive agreement (PPA) or negative productive agreement (NPA) with the reference assay.

6.In this study, the two assays used for validation are 'the OT DETECTR' targeting 16S rRNA and nested PCR targeting the 56 kDa gene. Please discuss this issue further, especially in light of the findings that there are 10-fold differences in the sensitivity.

7.Line 223: “All O. tsutsugamushi cases were laboratory confirmed at Jeju National University Hospital (Jeju, Korea)”

-Please provide more detail on the laboratory confirmation process. For clinical samples, scrub typhus infection is typically confirmed using a combination of antibody detection (such as serological assays; IFA) and antigen detection methods (e.g., PCR or Nested PCR) or bacterial isolation.

8.Line 227: Nested PCR was performed as previously described to detect the 56-kDa antigen of O. tsutsugamushi in extracted bacterial DNA

-Nested PCR not detect 56-kDa antigen, but detect the gene encoding the 56-kDa antigen, please clarify/correct.

9.Line 275: (B) OT2. (n = 6 replicates). No “.” After OT2

**Results**

-Does the analysis presented match the analysis plan?

-Are the results clearly and completely presented?

-Are the figures (Tables, Images) of sufficient quality for clarity?

Reviewer #1: I would consider revising the statistical analysis. While the manuscript mentions the use of Student's t-test to evaluate differences between groups, the results lack thorough presentation of statistical significance directly within the figures and text. Important statistical details, such as confidence intervals and exact p-values, are not consistently provided, which weakens the impact of the findings. In some figures (e.g., Figures 2, 5, and 6), statistically significant differences are not marked, making it difficult for readers to quickly assess the reliability of the data.

What is the point of the sensitivity comparisons? The sensitivity comparison between DNA and RNA targets (e.g., in Figure 2) could be more clearly explained. While it is mentioned that RNA-based detection is more sensitive, the reasons for this improved sensitivity are not fully explored, and the implications for clinical applications are not discussed in sufficient detail. This omission leaves readers without a clear understanding of the practical significance of these findings.

The results for clinical samples, presented in Table 1, lack a breakdown of potential variability in patient presentations or sample quality. Issues such as days of illness and days of presentation are not considered – please provide this information in the revision of the manuscript. The manuscript does not explore whether the one false negative result was related to sample handling, disease stage, or other clinical factors, which limits the interpretation of the test’s real-world applicability.

The figures and tables in the manuscript could benefit from some enhancements to improve their clarity and accessibility. Specifically, Figure 2, which compares the sensitivity of the OT DETECTR to Orientia tsutsugamushi DNA and RNA, has small labels and data points that are difficult to read. Increasing the font size of axes labels and improving the color contrast between data points would enhance its legibility. Similarly, Figures 5 and 6, which present the clinical applicability of the two-pot and one-pot DETECTR assays, could be improved by adding annotations to indicate statistically significant differences directly on the graphs, making the results more immediately clear.

Reviewer #2: (No Response)

Reviewer #3: Patient #27 Analysis: Could you expand on the potential reasons for missing one patient (Patient #27)? Was there something unique about this case, such as a low bacterial load, that could impact the sensitivity of the assay?

Reviewer #4: 1. In the plots, for Y axis labeled “Fluorescence” please specify the units of detection.

2.In the plots, for X axis suggest RNA copies/reaction

3.Do you have information on the O. tsutsugamushi genotypes in confirmed scrub typhus cases? Do you have a plan to perform genotyping on these samples?

4. Have you compared this newly developed assay with quantitative detection methods (such as qPCR assay in unit of copies/ml)? If not, please consider discussing how this comparison could enhance its applicability in clinical cases.

5.Have you statistically compare the fluorescent intensity obtained from two-Pot versus one-Pot approach?

6.Could you provide details on demographic background and/or clinical sings of the scrub typhus confirmed cases used in the study?

7.Line 314: “nested PCR” is a gold-standard test for diagnosing scrub typhus.

-Do you have any reference for the use of nested PCR as a gold standard for diagnosing scrub typhus

8.Line 348: Italic for O. tsutsugamushi

**Conclusions**

-Are the conclusions supported by the data presented?

-Are the limitations of analysis clearly described?

-Do the authors discuss how these data can be helpful to advance our understanding of the topic under study?

-Is public health relevance addressed?

Reviewer #1: The discussion and general conclusion in the manuscript present a promising outlook on the CRISPR-Cas12a-based diagnostic tool for scrub typhus, but the limitations are not sufficiently detailed or critically examined. While the authors emphasize the potential for rapid detection and field applicability, they do not adequately address key challenges such as the test's reliance on RNA extraction, which is difficult to manage in low-resource settings. Additionally, the limitations of the small sample size and lack of field testing are underplayed, which could affect the generalizability of the findings to real-world conditions.

Furthermore, while the discussion briefly acknowledges the need for improved nucleic acid extraction and portability, the authors should have provided a more detailed exploration of how these technological barriers might impact the test's practicality in resource-limited settings. The conclusion, though optimistic, could be more balanced by discussing these limitations in greater depth and suggesting clear pathways for overcoming them, such as developing more robust, simplified sample processing methods or conducting larger field-based trials. Addressing these gaps would make the conclusion more grounded and credible.

Reviewer #2: (No Response)

Reviewer #3: The Discussion does a great job summarizing the study’s key findings, comparing OT DETECTR to existing methods, and outlining future directions for diagnostic development. However, it would be beneficial to address the limitations of OT DETECTR in this section.

Reviewer #4: 1.Using the phase “Data not shown” can reduce transparency of the discussion suggest to avoid data not shown statement in the manuscript.

2.Please consider discussing the cost evaluation of the CRISPR-Cas12a-based POCT in comparison to the currently used diagnostic processes in hospitals

**Editorial and Data Presentation Modifications?**

Reviewer #1: While the manuscript is generally well-written, focusing on improving sentence clarity, streamlining some repetitive sections, and ensuring consistent formatting will improve its editorial quality. Proofreading for minor grammatical issues and enhancing the flow in certain areas will make it more reader-friendly.

It is unclear why the tables are clearly out of order - please re-order them so they read sequentially.

Please provide page numbers in the next revision.

Reviewer #2: (No Response)

Reviewer #3: 1.Line 48-49: Change "kDA" to "kDa."

Reviewer #4: Minor revision

**Summary and General Comments**

Reviewer #1: While the CRISPR-Cas12a-based OT DETECTR method shows potential in a controlled laboratory environment, its application in real-world clinical and field settings faces several challenges. These include lower sensitivity compared to nested PCR, technical difficulties related to RNA extraction and handling, and the need for further validation in diverse clinical samples. To be viable as a widely adopted diagnostic tool for scrub typhus, these issues must be addressed, particularly to ensure its reliability in resource-limited environments where the disease is prevalent.

Reviewer #2: The manuscript presents a CRISPR-Cas12a-based diagnostic method for detecting Orientia tsutsugamushi, the causative agent of scrub typhus. The work is interesting and provides a valuable tool for rapid diagnosis.

Major Comments:

Logical Structure of the Introduction: The introduction lacks a clear flow of ideas. It directly delves into the biological details of O. tsutsugamushi without first establishing the background and the need for rapid diagnosis. I recommend reorganizing the introduction to first address the clinical significance of scrub typhus and the challenges posed by current diagnostic methods. Then, introduce the biological characteristics of the pathogen and explain why 16S rRNA was chosen as the detection target. The current discussion of antigen variability does not provide sufficient justification for the research focus of this work.

Overemphasis on Historical Background: The historical context of scrub typhus occupies a significant portion of the introduction, without directly contributing to the study’s objectives. I suggest simplifying or removing much of the historical background to better maintain focus on the diagnostic challenges and the novel aspects of this study.

Specificity Testing: The specificity validation in the manuscript is not comprehensive, as no tests are included against pathogens closely related to O. tsutsugamushi. To strengthen the reliability of the specificity data, I recommend including additional tests with related bacterial species. This would support the claim of high specificity and reduce potential concerns about cross-reactivity.

Reviewer #3: The authors have developed a CRISPR-Cas12a-based diagnostic method to detect multiple genotypes of O. tsutsugamushi by targeting the bacterial 16S rRNA. This method has been widely utilized for pathogen diagnosis, particularly due to its low cost, minimal instrument requirements, and reduced need for technical expertise. The OT DETECTR assay will be particularly valuable in field and low-resource laboratory settings, as it demonstrates high sensitivity and specificity.

The authors provide a thorough and clear background on Orientia tsutsugamushi. However, it would be beneficial to briefly elaborate on how antigenic variability impacts diagnosis and why this diversity complicates traditional diagnostic methods. This addition could strengthen the rationale for the use of molecular diagnostics in this context.

Reviewer #4: This manuscript is overall well-written and presents a promising new method that could be a valuable tool for scrub typhus diagnostics. However, there are areas that could benefit from improvement as mentioned above.

PLOS authors have the option to publish the peer review history of their article (what does this mean?). If published, this will include your full peer review and any attached files.

Reviewer #1: No

Reviewer #2: No

Reviewer #3: **Yes: **Piyanate Sunyakumthorn

Reviewer #4: No

---

## [Decision Letter · Decision Letter 1]

6 Jan 2025

Dear Dr. Song,

We are pleased to inform you that your manuscript 'A CRISPR-Cas12a-based universal rapid scrub typhus diagnostic method targeting 16s rRNA of Orientia tsutsugamushi' has been provisionally accepted for publication in PLOS Neglected Tropical Diseases.

Best regards,

Yong Qi

Academic Editor

Georgios Pappas

Section Editor

Shaden Kamhawi

co-Editor-in-Chief

Paul Brindley

co-Editor-in-Chief

Reviewer's Responses to Questions

**Key Review Criteria Required for Acceptance?**

**Methods**

-Are the objectives of the study clearly articulated with a clear testable hypothesis stated?

-Is the study design appropriate to address the stated objectives?

-Is the population clearly described and appropriate for the hypothesis being tested?

-Is the sample size sufficient to ensure adequate power to address the hypothesis being tested?

-Were correct statistical analysis used to support conclusions?

-Are there concerns about ethical or regulatory requirements being met?

Reviewer #1: (No Response)

Reviewer #2: (No Response)

Reviewer #4: It's great to include the additional assay validation for Rickettsia spp. organisms.

**Results**

-Does the analysis presented match the analysis plan?

-Are the results clearly and completely presented?

-Are the figures (Tables, Images) of sufficient quality for clarity?

Reviewer #1: (No Response)

Reviewer #2: (No Response)

Reviewer #4: All my comments have been adequately addressed.

**Conclusions**

-Are the conclusions supported by the data presented?

-Are the limitations of analysis clearly described?

-Do the authors discuss how these data can be helpful to advance our understanding of the topic under study?

-Is public health relevance addressed?

Reviewer #1: (No Response)

Reviewer #2: (No Response)

Reviewer #4: All my comments have been adequately addressed.

**Editorial and Data Presentation Modifications?**

Reviewer #1: (No Response)

Reviewer #2: (No Response)

Reviewer #4: Accept the revised version with no further comments.

**Summary and General Comments**

Reviewer #1: (No Response)

Reviewer #2: (No Response)

Reviewer #4: The revised version has been further refined and thoroughly addresses all of my feedback.

I have no further comments.

PLOS authors have the option to publish the peer review history of their article (what does this mean?). If published, this will include your full peer review and any attached files.

Reviewer #1: No

Reviewer #2: No

Reviewer #4: No

---

## [Editor Report · Acceptance letter]

17 Jan 2025

Dear Dr. Song,

We are delighted to inform you that your manuscript, "A CRISPR-Cas12a-based universal rapid scrub typhus diagnostic method targeting 16S rRNA of *Orientia tsutsugamushi*," has been formally accepted for publication in PLOS Neglected Tropical Diseases.

Best regards,

Shaden Kamhawi

co-Editor-in-Chief

Paul Brindley

co-Editor-in-Chief
